# Stick around: Cell–Cell Adhesion Molecules during Neocortical Development

**DOI:** 10.3390/cells10010118

**Published:** 2021-01-10

**Authors:** David de Agustín-Durán, Isabel Mateos-White, Jaime Fabra-Beser, Cristina Gil-Sanz

**Affiliations:** Neural Development Laboratory, Instituto Universitario de Biomedicina y Biotecnología (BIOTECMED) and Departamento de Biología Celular, Facultat de Biología, Universidad de Valencia, 46100 Burjassot, Spain; david.deagustin@uv.es (D.d.A.-D.); isabel.mateos@uv.es (I.M.-W.); jaime.fabra@uv.es (J.F.-B.)

**Keywords:** CAMs, classical cadherins, nectins, neocortical development, radial glia cells, neurons, neuronal migration, axon targeting, synaptogenesis, neurodevelopmental disorders

## Abstract

The neocortex is an exquisitely organized structure achieved through complex cellular processes from the generation of neural cells to their integration into cortical circuits after complex migration processes. During this long journey, neural cells need to establish and release adhesive interactions through cell surface receptors known as cell adhesion molecules (CAMs). Several types of CAMs have been described regulating different aspects of neurodevelopment. Whereas some of them mediate interactions with the extracellular matrix, others allow contact with additional cells. In this review, we will focus on the role of two important families of cell–cell adhesion molecules (C-CAMs), classical cadherins and nectins, as well as in their effectors, in the control of fundamental processes related with corticogenesis, with special attention in the cooperative actions among the two families of C-CAMs.

## 1. Introduction

The nervous system structure and functionality are sustained by an exceptionally complex network of interconnected cells. This depends on the controlled spatial and temporal expression of selective molecules mediating various forms of contacts between different neural cell surfaces [1], the so-called cell adhesion molecules (CAMs). Since the 1970s, several families of CAMs have been described [2], as well as their physiological and pathological roles in the formation of the different structures conforming the nervous system, including the neocortex. This region contains hundreds of different cell types assembled into neural circuits dedicated to complex tasks such as cognition, reasoning, language or sensory perception, among others [3]. Therefore, the formation and maintenance of specific and precise cell contacts are critical for the proper functioning of cortical cells [4], where CAMs display key functions providing a platform for these cell–cell interacting processes [5]. Specifically, in the context of neurodevelopment, CAMs are involved in the control of progenitor behavior, neural migration, axonal pathfinding and wiring, along with synapse formation [5,6,7,8,9]. In fact, alterations of these proteins have been extensively related to a great number of neuropsychiatric disorders [10,11].

For attempting these functions, CAMs are located at the plasmatic membrane and exhibit unique structural features. Mostly as transmembrane proteins, these molecules possess distinct intracellular domains that can bind cytoskeletal as well as signaling partners that even have enzymatic activities [11]. All of these facts highlight the importance of CAMs also as modulators of intracellular events that ultimately affect cell behavior. In their extracellular regions, these proteins display numerous conserved motifs responsible for different adhesion modalities, thus increasing the complexity of the established contacts. Depending on the nature of the adhesive interfaces engaged in the interaction, CAMs can be broadly divided into cell–matrix and cell–cell interacting molecules. The first group comprises receptor molecules that mediate the binding of cell surfaces and different ligands at the extracellular matrix (ECM) [1]. Integrins constitute one of the main families of cell–matrix adhesion molecules and numerous studies have identified several relevant roles in different events within the development of the nervous system [12,13,14,15]. The second group of CAMs is composed by proteins that enable cell–cell interactions. According to their different extracellular domains, they have been classified into distinct superfamilies, including cadherin and immunoglobulin (IgSF), which in turn comprises proteins like NCAMs, L1, and nectins [10,11,16,17]. Importantly, some cell–cell adhesion molecules (C-CAMs) can interact either homophilically and/or heterophilically in opposite cells (termed as “trans-interactions”) or in the same cells (“cis-interactions”) [2,11,18,19,20,21]. This highly diverse recognition selectivity can be even increased by alternative splicing [1] and by post-translational modifications, such as glycosylation and phosphorylation [11]. Additionally, these proteins can provide specificity by their expression patterns in time and space [22,23,24,25].

Although the roles of both cell–matrix and cell–cell adhesion molecules have been widely recognized within neural development, this review will focus on C-CAMs and, more specifically, on classical cadherin and nectin families, with special interest in their cooperative behaviour regulating different processes throughout mammalian neocorticogenesis. The main properties of these molecules will be first examined to later discuss their implications in such mechanisms. 

## 2. Classical Cadherins

Both in simple and complex organisms, cadherins mediate Ca^2+^-dependent cell–cell adhesion and recognition through cis- and trans-interactions mediated by highly conserved cadherin-specific β-fold ectodomains (ECs) tandemly disposed in their extracellular regions, termed cadherin repeats (Figure 1A) [1,26,27,28,29]. Within the boundaries between these ECs, three consensus Ca^2+^-binding sites are displayed [1,30]. In the case of classical cadherins, only the members of type I/II subfamily are expressed in mammals [1,31,32]. Albeit some of them can also form weaker heterophilic interactions, type I cadherins (Cdh1, 2, 3, 4 and 15) mainly mediate strong homotypic cell–cell adhesion [5,24,28] (Table 1). Conversely, members of the type II subfamily (Cdh5-12, 18-20, 22 and 24) exhibit less robust homophilic contacts and heterotypic ones are more frequently observed among them [5,32]. In all cases, both trans and cis interactions among cadherins are essential for their adhesive function. 

In addition to ECs, type I/II classical cadherins possess a transmembrane domain and a highly conserved intracellular region containing distinct cytoplasmic domains, including a juxtamembrane domain (JMD) and a catenin-binding domain (CBD), that strengthen cell–cell adhesion and determine the effects on cell behavior [1,28,32,33] (Figure 1A). Catenins provide the anchorage between cadherins and the actin cytoskeleton, essential for cadherin-mediated cell–cell adhesion [34], and can be classified into the α-, β-, and p120-catenin groups [5,33] (Figure 1B). Specifically, αE/N-catenin displays characteristic vinculin/α-actinin- and ZO-1-binding domains and link the actin cytoskeleton to the β-catenins [5]. In turn, β-catenins bind to the CBD through several Armadillo repeats and p120-catenin to the JMD, the latter is thought to regulate cadherin surface stability and to modulate actin assembly [35,36,37]. 

Indeed, the existence of a crosstalk between cadherins and other cellular systems is widely acknowledged, such as FGF and EGF receptors, Wnt signaling, Nf-kB, Ras, and Hh signaling, which highlights the diverse roles of cell–cell adhesion as a regulator of cellular behavior [5,38]. 

Importantly, classical cadherins have distinctive and dynamic expression patterns throughout neural development in mammals [5,24,34], suggesting their key relevance in the regulation of specific processes in particular locations. While type I cadherins typically acquire broad distributions that are segregated by embryonic germ layer or tissue type [39], type II cadherins exhibit more fine-grained and often overlapping patterns of expression within individual cells and tissues [40,41].

## 3. Nectins and Nectin-Like Molecules

Nectins and nectin-like molecules (Necls) are immunoglobulin-like cell adhesion proteins (IgCAMs) that mediate Ca^2+^-independent cell–cell adhesion in several tissues during development [6]. The nectin family is composed of four members: nectin1, nectin2, nectin3, and nectin4, and all of them present two or three splice variants [42]. All nectins share similar structural features: an extracellular region distinguished by three Ig-like loops (one V-type and two C2-type domains), a single transmembrane domain, and a cytoplasmatic tail (Figure 1C), except for the secreted protein nectin1γ which lacks a transmembrane domain [43]. All nectins, besides nectin1β, 3γ, and 4, contain a conserved motif at their cytoplasmic tail (Glu/Ala-X-Tyr-Val), which serves as binding motif for the PDZ domain present in the scaffolding actin-binding protein afadin [6,44,45,46,47]. Despite the lack of this conserved motif, nectin4 binds the PDZ domain of afadin by its C-terminal motif (Gly-His-Leu-Val) [47]. 

Afadin presents two main splice variants: l-afadin and s-afadin [44,48]. l-Afadin, the longest, contains an F-actin binding site, binding F-actin along its sides and showing F-actin crosslinking activity. Afadin contains other domains: two Ras-associated domains (RA), a forkhead-associated domain (FHA), a dilute domain (DIL), a PDZ domain, and three proline-rich domains (PR), listed from N- to C-terminus [43] (Figure 1D). These domains enable the binding of several protein partners, among them, nectins are its most important interacting molecules serving as linkage to the actin cytoskeleton. In addition to afadin, nectins can bind other peripheral membrane proteins by their cytosolic tails, including the cell polarity protein Par-3 [49,50]. 

The extracellular regions of nectins are known to trans-interact homophilically and heterophilically with each other to mediate cell–cell adhesion [45,46]. First, homo-cis-dimers are formed, which experience lateral clustering on the plasmatic membrane. Then, homo- and hetero-trans-dimers are established between homo-cis-dimers clusters of opposing cells promoting cell–cell adhesion [46,51]. Unlike cadherins, nectins form stronger heterophilic than homophilic interactions among their family members [52]. This property is a key feature for their unique functions enabling the adhesion between heterotypic cells which express different nectin family members [53]. Moreover, the extracellular region of nectins is known to form heterophilic interactions with other proteins outside its family, promiscuously binding Necl and other IgCAMs, such as tactile, DNAM-1, and TIGIT [43]. Among nectin trans-interactions, nectin1/nectin3 is the strongest (K_d_ = 2.3 nM) followed by nectin2/nectin3 interaction (K_d_ = 360 nM) [52]. 

The Necl family is formed by five members: Necl1, Necl2, Necl3, Necl4, and Necl5. Necl share the same domain architecture as nectins yet lacking the afadin-binding conserved motif at their cytoplasmic tail (Figure 1C) [54]. Despite their inability to bind afadin, Necls still interact with multiple scaffolding proteins and participate in cell–cell adhesion [6]. Necls present multiple intracellular binding domains such as PDZ domain, FERM domain, and the protein 4.1 binding motif [43]. Membrane-associated guanylate kinase family members like Pals2 or Dlg3/MPP3 bind to the PDZ domain of necl1-2. Necl-2 additionally interacts, through the 4.1 domain, with the tumor suppressor gene product DAL1 [6]. As well as nectins, Necl extracellular region interacts to form homo-cis-dimers in the same cell surface and homo- and hetero-trans-dimers with interacting cells. All Necls can form homophilic and heterophilic interactions among nectin and Necl family members in a wide variety of combinations, but necl3 and necl5 are unable to form homodimers [55,56,57,58,59,60] (Table 1). 

Although the expression of nectins and Necls in the brain has not been as extensively characterized as classical cadherin presence, several publications have described their expression in different brain regions including the hippocampus [61] and the neocortex [22], where they play important roles regulating adhesive interactions.

## 4. Cadherin and Nectin Roles during Mammalian Neocorticogenesis

While cadherins and nectins have well-characterized independent actions, research over the past few decades have also unveiled their great collaborative capacities. As a matter of fact, nectins cooperate with cadherins in the assembly and maintenance of specialized contact zones between epithelial cells, called adherens junctions (AJs) [43,45]. These AJs are dynamic structures regulated by the actin cytoskeleton and whose formation is initiated by the accumulation of nectins on the surfaces of two apposed cells [34,43]. Nectin trans-interactions and the subsequent assistance of adaptor proteins such as afadin [62,63] and catenins [33] enable the recruitment and clustering of classical cadherins to the adhesion sites, stabilizing the junctions. This synergism between classical cadherins and nectins appears in several processes throughout neocorticogenesis where cell–cell interactions are required. Because of this reason, the next sections will concomitantly cover and examine the roles of these C-CAMs in such cellular events.

### 4.1. RGCs Maintenance, Proliferation, and Fate Determination

Before the onset of cortical neurogenesis (E10.5–E11.5 in rodents), the neuroepithelial cells located in the neural tube transform into radial glial cells (RGCs), which are the cortical neural stem cells [64,65]. Importantly, RGCs situated in the pallial germinal zone, also known as the ventricular zone (VZ), will give rise directly or indirectly to all the excitatory cortical projection neurons as well as macroglia cells at later stages that will eventually conform the six-layered neocortex, while RGCs restricted to the subpallial ganglionic eminences will initiate cortical interneurons [3,66,67,68]. 

Cortical RGCs mostly divide symmetrically at early embryonic ages to amplify the stem cell pool, but over time RGCs tend to divide asymmetrically to self-renew and to generate a non-radial glial daughter cell, which can be a postmitotic cell or another type of progenitor cell known as an intermediate progenitor (IP). IPs exit the VZ to locate in more basal positions and form the subventricular zone (SVZ) [69,70]. Some IPs divide symmetrically a limited number of times to produce new IPs before giving rise to neurons, whereas other IPs directly undergo symmetrical terminal divisions to generate pairs of neurons. Nevertheless, they are additional types of progenitors, such as short neural progenitors (SNPs) [71] and outer radial glia cells (oRGCs) [72,73,74,75]. SPNs are located in the VZ and originate from RGCs [71]. In contrast to RGCs, they lack a basal process that reaches the pial surface, but their apical process maintains contact with the ventricular surface. oRGCs are also generated from RGCs and retain the basal (pial) process but not the apical one. In animals with expanded cortices, oRGCs primarily populate a subdivision of the SVZ known as the outer subventricular zone (oSVZ). Although they were originally identified in gyrencephalic animals, oRGCs have been also described in rodent brains and lissencephalic primates [76,77]. 

A substantial number of studies have examined the implication of C-CAMs in the control of neural progenitor behavior and how the adequate organization and maintenance of neuroepithelial AJs are essential for neocortical RGCs. At this location, AJs provide stability to maintain tissue cohesion but also offer enough plasticity to allow cellular rearrangements during neurodevelopment. The perturbation of these cell–cell contacts causes VZ disruption, loss of normal apicobasal polarity, and detachment of RGCs from the apical surface. Such alterations in RGCs result in important cell autonomous and non-cell autonomous defects during cortical development, ranging from changes in the division type to lamination problems [78,79,80,81]. Thus, from the study of conditional knockout mice, it is known that AJs formation and maintenance between RGCs in the developing neocortex rely on several junctional proteins, including Cdh2, αE-catenin, β-catenin, and afadin [79,80,82,83] (Table 2) (Figure 2). Interestingly, perturbation of αE-catenin [83], Cdh2, and afadin [80] since the early stages of cortical development increase the proliferation of progenitor cells causing cortical hyperplasia and several laminar alterations including double cortex formation (Table 2) (Figure 2A,B). However, the elimination of some of these junctional proteins in RGCs, like afadin and Cdh2, at later stages of cortical development produces incomplete disruption of AJs [84] or milder genotypes [79], highlighting the importance of early deletion of these proteins to disrupt the normal behavior RGCs (Table 2). 

The fact that the perturbation of several AJ-associated genes including Cdh2, afadin and αE-catenin is related to higher progenitor proliferation may suggest that disruption of AJs per se could be responsible for the over-proliferation phenotype. However, the deletion of other essential AJ-associated molecules, such as β-catenin, since the early stages of the developing neocortex also trigger a severe loss of AJs but decrease proliferation rates since E15.5 [82,88] (Table 2) (Figure 2A,C). In a similar way, apical-complex-protein Pals1 ablation using Cre-mediated recombination driven by Emx1 promoter disrupts AJs and happens to deplete the pool of neocortical progenitor cells inducing premature cell cycle exit, giving rise to an excessive number of early-born postmitotic neurons that eventually die [89]. 

All of this experimental evidence strongly points towards an intriguing question: is it possible that molecules that mediate cell–cell contacts further govern cell proliferation within the developing neocortex through downstream signaling mechanisms that are independent from their adhesive role? This would not mean that the loss of ventricular and subventricular zone cytoarchitecture upon the deletion of these genes does not have an influence on the cell division rate of progenitors, but that it could not be the only reason at the molecular level underlining enhanced proliferation. In this sense, it has been described that the aberrant progenitor proliferation observed in αE-catenin conditional mutants is actually mediated by an increased activation of the Hh signaling [83]. Widely recognized as one of the major molecular pathways in the context of neurodevelopment in general and corticogenesis in particular [90], Hh signaling appears to be crucial for the expansion of oRGCs [91] and more recent data have revealed differential Shh signaling in a subpopulation of oRGCs [92]. oRGCs have been described as key progenitors that contribute to the generation of upper-layer neocortical projection neurons in gyrencephalic mammals, including humans [93,94,95,96]. 

Very appealingly, cortical projection neurons expressing upper-layer markers seem to be more abundant in Cdh2 and afadin conditional mutants at postnatal ages, which indeed exhibit an enlarged neocortex [80]. Additionally, the lack of the apical process in oRGCs is a feature also displayed by progenitor cells whose proliferation is highly increased upon Cdh2 and afadin ablation in mice [80], and Cdh1 downregulation during the expansion period of oRGCs has been described in ferrets [95]. Such findings suggest an important function of these C-CAMs in the expansion of oRGCs. Hence, new experiments are required to be unveiled if, besides regulating cell–cell contacts and neocortical progenitor proliferation, Cdh2, afadin, and other related proteins could govern to some extent cell fate decisions at the RGC level and whether this regulation takes place through the control of oRGCs generation. The fact that higher amounts of II/III-layer neocortical projection neurons have been correlated with the existence of autism-like features in mice [97] makes these data even more fascinating, as it could help to provide basic knowledge that may improve our understanding of the molecular and cellular basis underlying neurodevelopmental disorders. Together with the previous mentioned work, these investigations emphasize the role of C-CAMs and their related proteins expressed in cortical progenitors within the regulation of stem cell maintenance, proliferation, neuronal differentiation, and cell fate choice in the context of mammalian neocorticogenesis. 

### 4.2. Neuronal Migration

The birthplace of postmitotic neurons, generated by the previously mentioned neural progenitor cells, differs from the destination place within the cortical plate where they will be assembled into neural circuits. The C-CAMs participate in this process mediating cell–cell interactions required for neural cells movement. Defects in the correct cell positioning are responsible for several neurodevelopmental disorders, such as lissencephaly and focal cortical dysplasia [98]. Because of the complex and laminated structure of the neocortex, postmitotic neurons employ two different modes of migration according to their birthplace. Cortical projection neurons migrate radially from the ventricular and subventricular zones (VZ and SVZ) to reach their target position in the incipient cortical plate [3,99,100]. However, cortical interneurons, generated in the ganglionic eminences, travel long distances across tangential streams and end their migration by radial movement once in the cortical plate [100,101,102]. Not only the cortical interneurons migrate tangentially, but Cajal–Retzius (CR) cells, a pioneer cortical neuronal population, originating at different locations outside of the cortical proliferative region [103,104], also reaches their superficial locations at the marginal zone (MZ) by this mode of migration. Once there, CR cells guide the radial migration of projection neurons and coordinate cortical lamination [105,106].

At early stages of cortical neurogenesis (E11.5–E13.5), postmitotic cortical projection neurons can extend and anchor their leading processes into the marginal zone (MZ) and translocate their soma to be placed in the preplate in a process named “glia-independent somal translocation” (Figure 3A) [107]. These early-born neurons give rise to the deep neocortical layers (V–VI) also known as “lower-layer” neurons [3,108]. On the other hand, the “upper-layers” neurons, born later (E14.5–E.16.5), populate the most superficial layers (II–IV). Due to the cortical thickness at this developmental time, projection neurons need to use several sequential modes of radial migration through the different areas of the cortical primordium. In the first stage, recently generated postmitotic neurons detach from the neuroepithelium and progress radially to the SVZ [70]. In the SVZ, postmitotic neurons present indefinite polarity, emitting and retracting extensions in order to get to the intermediate zone (IZ) (Figure 3A). This type of cell motility is known as “multipolar migration” [109]. Once in the IZ, the migration mode used by migrating neurons is the “glia-dependent locomotion” [110]. Neurons acquire bipolar morphology, extend a short leading process, and translocate the nucleus, as well as the rest of the cell body moving along RGCs fibers describing a saltatory pattern (Figure 3A). Finally, once the neuronal leading process can be anchored to the MZ, these neurons complete their migration by glia-independent somal translocation [100,107]. To some extent, C-CAMs have been found to participate in the different migration modes described for projection neurons.

Somal translocation and multipolar migration are processes orchestrated by reelin, a glycoprotein secreted by CR cells [111,112,113]. The absence of reelin during corticogenesis, studied in a reelin-deficient mouse (reeler mutant), causes severe neocortical disorganization [114,115]. Reelin acts via its receptors (VLDLR and ApoER2) present in both RGCs and migrating neurons [116,117,118,119]. Reelin binding to its receptors leads to the phosphorylation of intracellular adaptor Dab1 by Src-family kinases (Src and Fyn) [120,121]. Phosphorylated Dab1 then recruits PI3K [122] and Crk/Crkl/CG3 [117,123], as well as activates some important downstream effectors, such as Limk1, Akt, and Rap1, that regulate the actin cytoskeleton and the neuronal adhesive properties, among other functions [123]. 

In the specific case of multipolar migration, the presence of reelin in the IZ allows for migrating neurons to orient toward the cortical plate by the activation of the small GTPase Ras-related protein 1 (Rap1) [124,125]. This Ras-GTPase increases Cdh2-surface levels, promoting the transition to bipolar morphology required for glia-dependent locomotion [124,126]. Likewise, reelin-Dab1-Rap1 signaling has been described to encourage the somal translocation of early- and late-born neurons through Cdh2-homophilic interactions [127]. Dab1 would be responsible for leading-process stabilization by Rap1-regulation, which in turn will control cadherin function to conduct somal translocation (Figure 3B). Dab1 loss in postmitotic neurons results in cell autonomous deficits and a subsequent phenotype similar to the reeler mutant [127].

Cadherins are not the only C-CAMs that support somal translocation; a whole combinatorial code of adhesion molecules between CR cells and migrating neurons has been described in this process [22]. The IgCAM nectin1 is specifically expressed in CR cells at early embryonic ages, while nectin3 is present in postmitotic migrating neurons. The heterophilic interaction of both nectins is just as necessary as the function of their intracellular effector afadin to make the leading process anchor within the MZ. Afadin binds Rap1 to recruit p120-catenin, a critical regulator of cadherin stability in migrating neurons. Therefore, the reelin-Dab1 signaling activates Rap1 [127] at the same time that interactions between nectins through afadin make the Cdh2 recruitment possible [22] (Figure 3B). Cdh2 molecules are expressed ubiquitously in both cell types, and the strengthening of their homophilic interactions allows the stabilization of the initial nectin-based contacts to allow neuronal somal translocation once the leading process is anchored in the MZ. This cooperative adhesive mechanism was previously described in AJ formation [45].

Glia-dependent locomotion seems to be independent of reelin-signaling [124,127] but not of cadherin function. Late-born migrating neurons need to establish dynamic connections to travel along the fibers of RGCs [110] (Figure 3C). A sophisticated and coordinated mechanism of different endocytic pathways mediated by several members of the Rab-GTPase family, including Rab5 and Rab11, has been reported to mediate this type of migration through the regulation of Cdh2 trafficking toward the distal tip of neuronal leading processes [128] (Figure 3C′). Recently, other studies have proposed additional pathways related with this Cdh2-trafficking. This is the case of the adaptor protein Debrin-like (Dbnl), that acts promoting Cdh2 expression on the surface of migrating neurons [129] and the ADP-ribosylation factor 6 (Arf6), an additional small GTPase that regulates Cdh2-endosomal recycling through Rab11 family interacting protein 3 (FIP3) [130]. In addition, another relevant aspect of the locomotion migration is the nucleokinesis, which needs precise control of the cytoskeleton to happen. Cdh2 and Cdh4 cooperate homo- and heterophilically to regulate locomotion during glia-guided migration by promoting interactions among migrating neurons and RGCs [19]. This regulation occurs via protein-tyrosine phosphatase PTP1B and α/β-catenins, connecting cadherins with the actin cytoskeleton (Figure 3C″). A possible interaction of Cdh2 with microtubules through Lis1 has also been described [19]. According to these data, cadherins could act providing the leading processes of migrating neurons the necessary traction to convert contractile forces into forward-movements of the nucleus.

Apart from radial migration, C-CAMs have also been related with the control of tangential migration of interneurons. Specifically, Cdh2 expression has been described as necessary to promote cell motility and keeping cell polarity in postmitotic neurons during the whole process [131]. Experiments disturbing Cdh2 expression in the medial ganglionic eminence cells lead to cell autonomous deficits and polarity changes that blocked tangential migration. Problems in the cortical plate invasion also occurred upon Cdh2 ablation in migratory interneurons. The polarity alterations were associated with the regulation of centrosome positioning, actomyosin-contractile cytoplasmic distribution, and leading process stabilization [131]. Additional work is necessary to gain more insights into the possible roles of additional cadherins and nectins in the regulation of this mode of neuronal migration within the cortical plate.

### 4.3. Axonal Outgrowth and Target Recognition

During migration to their places of destination in the cortical plate, projection neurons extend their axons to contact specific target cells and establish synaptic contacts [70,132] (Figure 4A). Considering the highly organized structure of neocortex in layers, columns, and areas, the existence of accurate mechanisms synchronizing axonal pathfinding and assuring proper target recognition are of crucial importance. A diversity of attractive and repulsive extracellular guidance cues helps the axonal growth cone to navigate through the developing brain. The nature of these cues is wide: from secreted extracellular matrix proteins, such as netrins, slits, and semaphorins, to C-CAMs, such as ephrins, cadherins, and nectins [133,134]. Cortical axons of distinct neural populations selectively connect with specific target cells that may be located at short or far distances, including structures outside the neocortex. Furthermore, axons often elongate alongside other pioneer axons, and C-CAMs also take part in axon–axon interactions to axon bundling, also called fasciculation [6,135]. 

The implication of cadherins and nectins regulating axonal guidance and target recognition has been more deeply characterized in structures outside of the neocortex, including the spinal cord and motor or sensory nerves [135,136,137]. This is due, more likely, to the vast complexity of neuronal wiring in this region, which makes their study an authentical challenge. Nevertheless, different roles related with axonal pathfinding and target recognition have been described in different stages of cortical development. A recent study suggests that Cdh2 is involved in the very early stages of axonal pathfinding by regulating polarization and the initiation of axon outgrowth [126]. Something very interesting related with some of these C-CAMs, the classical cadherins, is how their combinatorial expression seems to correlate with the functional organization of neurons in circuits. This way cadherins could act providing adhesion codes useful for functional organization of neural connections [138]. Some examples of this combinatorial code have been found in layer IV of the primary somatosensory cortex (the barrel cortex), which contains an isomorphic map of the large facial whiskers of the contralateral snout. This barrel cortex receives thalamocortical inputs from two different thalamic nuclei, the ventral posterior medial nucleus (VPN) and the posterior nucleus (POm) in two mutually exclusive locations (barrel hollows vs. barrel septa). Histological studies have found that Cdh2 expression is concentrated in the VPN cells and their targets cells in the barrel hollows [139], whereas Cdh8 expression is concentrated in cells of the other pathway [140], suggesting that different types of cadherins regulate the particular connectivity between different thalamic nuclei and their target regions in the neocortex. Perturbation of Cdh2, using blocking antibodies in organotypic thalamic and cortical co-cultures, produces altered innervation patterns of thalamocortical axons that aberrantly target pial surface instead of layer IV [141], confirming the role of cadherins in the guidance of thalamocortical axons. Cadherins have also been observed to be involved in the regulation of one type of intracortical axonal connection, known as “barrel nets”, originated by layer II/III projection neurons and their targets in layer IV barrels of the primary sensory cortex [142]. These authors used dominant-negative strategies [19,22,127,128], perturbing the function of all classical cadherins, in layer II/III projection neurons, by in utero electroporation and observed that barrel nets were disrupted, whereas no defects in callosal projections were found. Nonetheless, de novo pathogenic variants in Cdh2, targeting EC4 and EC5 domains, cause neurodevelopmental disorders with defects in callosal projections from mild hypoplasia to complete agenesis, suggesting the implication of Cdh2 in the control of this type of axonal projections [143].

Another cadherin that also participates in axonal pathfinding is Cdh13 (T-cadherin), the only member of classical cadherins anchored to the membrane through a glycosylphosphatidylinositol moiety (GPI) [144]. Cdh13 lacks typical transmembrane and cytoplasmatic domains of classical cadherins and its expression is restricted to axons of a subset of subcerebral corticofugal projection neurons in late embryonic stages [145,146]. In addition to axonal guidance, the homophilic binding between Cdh13-expressing cells appears to help axonal fasciculation, as well as driving correct direction and orientation in the axons projecting to the internal capsule and the cerebral peduncle [146]. 

Although the involvement in axonal pathfinding and cell recognition has not been described to date for many cadherins or nectins in the neocortex, their particular laminar expression patterns reported by some authors [145,147,148,149] suggest that other C-CAMs, including nectin3 and Cdh22, specifically expressed in projection neurons of layers II/III and V, respectively, could be also involved in these processes. Further studies will need to be conducted in order to identify possible functions of these or other candidate genes as well as possible cooperative roles of nectins and cadherins mediating these modes of adhesive interactions.

### 4.4. Synapse Formation and Remodeling

The formation, elimination, and stabilization of functional synaptic connections is key to neural circuit wiring as well as proper brain development and function [86]. Among other C-CAMs, cadherins and nectins are of great importance as coordinated orchestrators of these processes [5,150,151,152,153]. Such roles have been widely studied in the CA3 area of the mouse hippocampus, where synapses are established between the terminals of mossy fibers and the dendrites of pyramidal neurons. In the recent years, the implication of C-CAMs in synapse formation and plasticity during neocortical development have begun to be examined and will constitute the particular interest of this section. 

Synapse formation relies on the clustering of adhesion molecules, signaling and scaffolding proteins and receptors into highly organized complexes [154,155]. It begins at late stages of embryonic development and mostly occurs postnatally through different steps, including neurite outgrowth, target recognition, synaptic maturation, and synapse remodeling [5,150,152,156,157] (Figure 4B). The involvement of cadherins and nectins at initial phases of neurite outgrowth has not received deep treatment in neocortical neurons. However, it has been demonstrated that Cdh8 expression in layer V-projection neurons of the medial prefrontal cortex is critical for their correct dendritic arborization at the dorsal striatum [158]. As this molecule is broadly expressed within the neocortex, it would be interesting to examine if its importance in dendrite formation is conserved among cortical layers. 

Upon target recognition, protruding dendrites are then able to start making primordial synaptic contacts with other neural structures, most commonly axonal boutons [5,153]. In the hippocampus, this selectivity is thought to be based on the heterophilic interaction between presynaptic nectin1 and postsynaptic nectin3 [5,151,159,160]. This enables the establishment of the first contacts between filopodia-protruding dendrites and axons of hippocampal neurons [5] (Figure 4B-1). Several nectin proteins, including nectin1 and nectin3, are also expressed in the adult murine neocortex [145,161], and a recent work has shown nectin3 overexpression in cortical RGCs since E15.5 postnatally disrupts synaptogenesis, whereas its knockdown enhances synaptic maturation [161]. Such effect is intriguingly the opposite than that observed in previous studies conducted in the hippocampus, where nectin1 in vitro inhibition [151] and nectin3 in vivo knockdown impair synapse formation and maturation [162,163]. Moreover, afadin has been identified as an important regulator of hippocampal synaptogenesis and synaptic function as its inactivation, since early neurogenesis, significantly reduces nectin1, nectin3, and Cdh2 presence at the synapses between mossy fibers and CA3 pyramidal cells [164]. Likewise, when afadin is ablated in postmitotic hippocampal neurons, Cdh2 is also reduced in the CA1 stratum radiatum together with αN-catenin, β-catenin, and excitatory synapse densities [165]. Given the involvement of afadin in hippocampal synaptogenesis and their known roles in other processes of corticogenesis, further studies are needed to characterize its implication at these initial stages of neocortical synapse formation.

In the hippocampus, after this nectin-dependent interaction, Cdh2 molecules are recruited in an afadin-dependent manner at both presynaptic and postsynaptic sides to establish homophilic interactions, subsequently strengthening primordial junctions between axons and the filopodia-protruding dendrites [5,6,166] (Figure 4B-2). This event is followed by the assembly of both presynaptic active zones and postsynaptic densities, at least partially driven by nectins1/3 and Cdh2 [6]. Nectin- and Cdh2-mediated cell adhesion is not only necessary for proper synaptogenesis, but also for synapse function and maintenance. Cooperation of nectin and cadherin pathways in synapse stabilization in the neocortex has not been described to date. Nevertheless, Cdh2-homophilic interactions in cortical synapses are important for their maintenance, as Cdh2 overexpression only in the post-synapsis of cultured neocortical neurons elicits axon retraction, impairs synaptic function, reduces presynaptic vesicle clusters, and eventually triggers synapse elimination in vitro [167]. Moreover, the presynaptic adhesive complex formed by this cadherin and catenins has been identified in vivo as a key stabilizer of functional excitatory synapses within layers II/III and IV of the postnatal developing somatosensory neocortex in mouse [86] (Table 2). In addition, a recent study ahead of publication indicates that the balance between Cdh8 and Cdh11 levels in neocortical and hippocampal neurons is essential for proper cortical connectivity and its alteration might be related to the etiology of autism [168]. These results suggest that interactions between different classical cadherins are crucially involved in synaptic development, and their participation could entail great relevance in pathological situations.

Once established, synapses mature by increasing their size and the number of synaptic proteins harbored at pre- and postsynaptic sides [157] (Figure 4B-3). In the case of most excitatory synapses within mammalian brain and neocortex, synaptogenesis is coupled to the transformation of dendrite’s filopodia into specialized and stabilized structures named synaptic or dendritic spines, where postsynaptic densities and associated organelles accumulate [5,169,170,171]. In rodents, rapid spinogenesis occurs during early postnatal stages short after synaptogenesis [169,172]. Afterwards, sensory experience-dependent synaptic activity elicits the maturation of part of these spines, while a significant number of them are eliminated by pruning to refine neural circuits during adolescence [152,161,169,170]. Dendritic spines can be further remodeled as a consequence of learning and memory acquisition [6,173]. Apart from having well-recognized roles in hippocampal spine formation and plasticity [5,6,174,175,176], some C-CAMs have also been found to be crucial for these events to take place in the neocortex. Recently, it has been demonstrated that nectin3 is involved in neocortical spinogenesis, as its in vivo overexpression or knockdown at E15.5 not only affects synaptogenesis, but also results in decreased and increased spinogenesis, respectively [161]. Besides nectin3, another member of the IgCAM superfamily, Necl2, has been recently reported as necessary for the maturation of the synapses between thalamocortical neurons and parvalbumin-positive interneurons within laver IV of the primary visual cortex, given that its ablation decreases the number of thalamic inputs received by these interneurons [87] (Table 2). In addition, the molecular complex formed by Cdh2 and β-catenin is necessary for an appropriate spine maturation and pruning within the postnatal somatosensory neocortex and hippocampus in vivo. In postmitotic PNs, either overexpression of the Cdh2-intracellular domain, sequestering β-catenin, or β-catenin deletion impairs such processes [85] (Table 2). Interestingly, locally enhancing the cell–cell adhesion mediated by the Cdh2/β-catenin complex, only in certain individual spines, provokes their maturation, whereas neighbor spines located in the same dendritic branch but displaying low levels of such complex are eliminated both in vitro and in vivo. Further in vivo analysis revealed that these complex drives activity-induced spine fate differentiation [85] and stabilization in layers II/III and IV of the somatosensory neocortex [86]. αN-catenin has also been found to be important for correct spine pruning and maturation of neocortical PNs, as its deletion impairs such processes [85] (Table 2).

Altogether, cadherin and nectins play critical roles in the formation, stabilization, and maturation of synaptic contacts. All these events are essential for proper brain function, and their perturbation has been related with several neurodevelopmental disorders such as autism, schizophrenia, or intellectual disability, among others [157,168,177,178,179]. Despite the fact that the implications of C-CAMs in excitatory synapse and spine formation are well understood thanks to the extensive research conducted in the hippocampus, new studies are required to continue unveiling their specific contribution to the establishment of neural excitatory and inhibitory connections in the precise context of neocortical development.

## 5. Relevance of Classical Cadherins, Nectins, and Necls in Human Neurodevelopmental Disorders

Throughout the previous sections of the present review, the important implication of classical cadherins and nectins in the regulation of neocorticogenesis has been recognized. Hence, their disfunctions have dramatic consequences and have been found to be responsible for several neurodevelopmental disorders [6,180,181]. Mutations in several C-CAMs have been detected in humans and associated mostly with Autism-Spectrum Disorder (ASD) [182,183,184,185,186,187], Attention-Deficit/Hyperactivity Disorder (ADHD) [184,188,189,190], Bipolar disorder (BD) [191], schizophrenia (SCZ) [192], learning disability (LD) [186], intellectual disability (ID) [143,193,194], obsessive-compulsive disorder (OCD), and Tourette Disorder (TD) [195], among others (Table 3). Nevertheless, to date, very few studies have been able to link these mutations in humans and the neurological disorders caused by them with specific neocortical malfunctions, given the difficulty to identify alterations in particular brain regions as responsible for concrete neuropsychiatric symptoms. Recently and as previously mentioned, in the case of the Cdh2 gene, seven missense substitutions affecting highly conserved residues in EC4/5 crucial for Ca^2+^-binding and impairing correct trans-interactions, together with two frameshift variations leading to a truncated cytoplasmic domain, have been described to cause corpus callosum agenesis or hypoplasia as well as intellectual disability in heterozygosity [143]. Substitutions in the Cdh4 gene have also been reported to entail corpus callosum dysgenesis, as well as microcephalia, in heterozygosity [196]. Neocortical alterations are also displayed because of microdeletions in the Cdh8 gene and are related to the susceptibility to ASD and LD [186]. Although mutations in members of the nectin family have not yet found to be associated with neuropsychiatric disorders, Necl2 has been associated by several studies, with ADHD and specifically with regional abnormalities in the right superior frontal gyrus of the neocortex [188]. All this evidence undoubtedly highlights the relevance of investigating the mechanisms in which classical cadherins, nectins, and the rest of the C-CAMs are involved not only in physiological circumstances, but also with how their alteration affects normal neurodevelopment of such an important area as the neocortex is. In this sense, further research will also need to uncover and dissect the affectation of brain functions as a consequence of the mutation of these genes in a regional- and process-specific manner.

## 6. Discussion

Neocortical development involves multiple cellular re-arrangements since neural progenitors originate and give rise to neural cells that delaminate from proliferative zones, migrate to their destination places and eventually find target cells to establish synaptic contacts. All of these complex events are tightly regulated because they are essential for sustaining proper brain function. In this context, cell surface adhesion molecules, including C-CAMs of classical cadherin and nectin families, display specific spatial and temporal expression patterns and have been described to cooperate in the regulation of those processes both through homophilic as well as heterophilic interactions [5,6]. For this cooperation to happen, adaptor proteins such as afadin and catenins are required for the nectins to recruit cadherins and establish different kinds of cell–cell contacts in terms of functional implications and adhesive strengths. Whereas in the neuroepithelium these C-CAMs are involved in the formation and maintenance of very stable AJs between neural progenitor cells, they are also able to establish highly dynamic and transient cell–cell junctions that are essential for neuronal migration by somal translocation. In this case, nectin heterophilic interactions between migrating cortical neurons and CR cells promote Cdh2 clustering to adhesion sites via afadin, Rap1, and p120 catenin to form homophilic interactions [22]. However, no cooperation between nectins and cadherins has been described during glia-dependent locomotion migration of cortical projection neurons, in which interactions between Cdh2 and Cdh4 can take place both homophilically and heterophilically [19]. Similarly, particular cadherin homophilic adhesion codes have been observed to mediate particular target recognition along axonal pathways, as found in the thalamocortical system [139,140]. In the hippocampus, after postmitotic neurons have reached their final locations, nectins and cadherins cooperate again in the formation of robust synapses [164]. Although this cooperative behavior, among both C-CAM families has not been described for synapse formation in the neocortex, it is possible to think that a similar cooperation could occur in this region. Additional work will need to be conducted to test this hypothesis.

Importantly, apart from mediating cell–cell adhesion during corticogenesis, C-CAMs such as Cdh2 and its related adaptor afadin act increasing proliferation of RGCs in an apparently adhesion-independent manner, as this phenotype is absent upon ablation of other junctional proteins [80,82,89]. Together with the observation of enlarged production of projection neurons expressing upper-layer markers in mutant mice for these genes [80], it is tempting to speculate that Cdh2 and afadin could govern downstream signaling pathways controlling the behavior of RGCs in terms of proliferation, differentiation, and cell fate choice. 

In addition to all the information about C-CAM roles regulating different processes of mammalian neocorticogenesis obtained from the analysis of different murine models, the relevance of those proteins during neurodevelopment is also known because mutations in many of these molecules have been found in patients of several neurodevelopmental disorders (Table 3). These data further suggest that correct functioning of C-CAMs is essential to maintain proper brain function. Nevertheless, and despite the great advances in the knowledge gained during the last two decades, many open questions still need to be elucidated. For example, what are the molecular mechanisms associated with the involvement of these junctional proteins in progenitor proliferation? Are these changes in proliferation affecting all RGCs or, instead, particular subpopulations of them perhaps through combinatorial adhesion codes mediated by differentially expressed cadherins or nectins? Is the differential expression of these C-CAMs in precise layers of the neocortex related with particular types of cortical connectivity? How many of the neurodevelopmental alterations found in patients displaying C-CAMs mutations are caused by specific neocortical malfunction? Additional functional studies will help to answer these questions so as to uncover the roles of C-CAMs in the control of neocorticogenesis and improve our understanding of the molecular and cellular alterations underlying several of the mentioned neurodevelopmental disorders.

## Figures and Tables

**Figure 1 cells-10-00118-f001:**
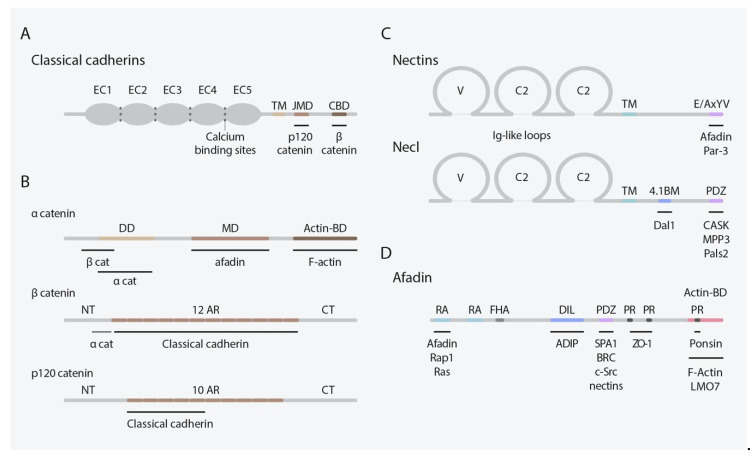
Classification of classical cadherins, nectins, Necls, and their principal binding partners. (**A**) Structural representation of type I/II classical cadherins. These cadherins contain an extracellular region to mediate adhesive interactions, a transmembrane domain, and a cytoplasmic tail with several binding domains to catenin proteins. (**B**) structural representation of catenin proteins. α-catenin can bind cadherins to the actin cytoskeleton through β-catenin. P120-catenin present a cadherin binding domain. (**C**) structural representation of nectin and Necl families. These Ig-CAMs contain three extracellular Ig-like loops necessary to establish adhesive contacts, a transmembrane region and a cytoplasmic region allowing the interaction with several binding partners including afadin. (**D**) structural representation of afadin displaying its multiple binding domains. Abbreviations: 4.1BM, 4.1 binding motif; actin-BD, actin-binding domain; AR, Armadillo repeats; C2, C2-type domain; CBD, catenin-binding domain; CT, C-terminal; DD, dimerization domain; DIL, dilute domain; EC, Ectodomain; FHA, forkhead-associated domain; JMD, juxtamembrane domain; MD, M-domain; NT, N-terminal; PDZ, PDZ domain; PR, proline-rich domain; TM, transmembrane domain; V, V-Type; RA, Ras-associate domain.

**Figure 2 cells-10-00118-f002:**
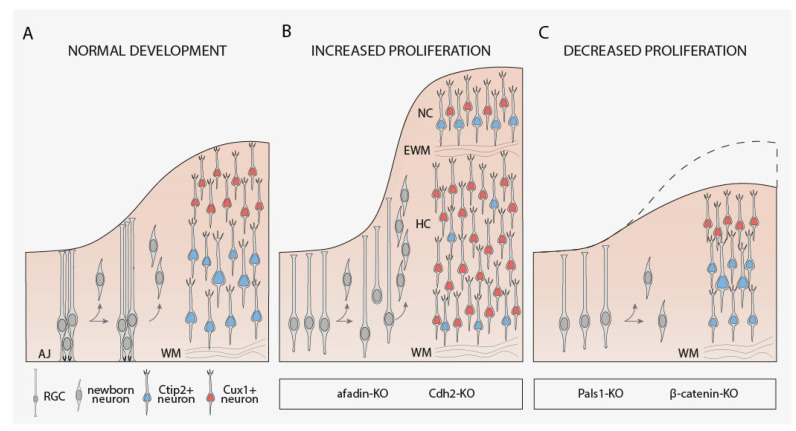
C-CAMs implication in control of proliferation during neocortical development. (**A**) schematic representation of the normal neocortex development. Radial glial cells (RGCs) mostly divide asymmetrically to self-renew and produce a postmitotic neuron or an intermediate progenitor cell (not shown). (**B**) perturbation of AJ-associated genes such as afadin and Cdh2 causes increased proliferation, cortical hyperplasia, double cortex formation, and enlarged production of Cux1+ neurons. (**C**) disruption of other junctional proteins like β-catenin and Pals-1 disassembles AJs causing a decrease in proliferation and leading to a reduced neocortical structure. EWM, ectopic white matter; HC, Heterotopic Cortex; NC, Normocortex; WM, white matter.

**Figure 3 cells-10-00118-f003:**
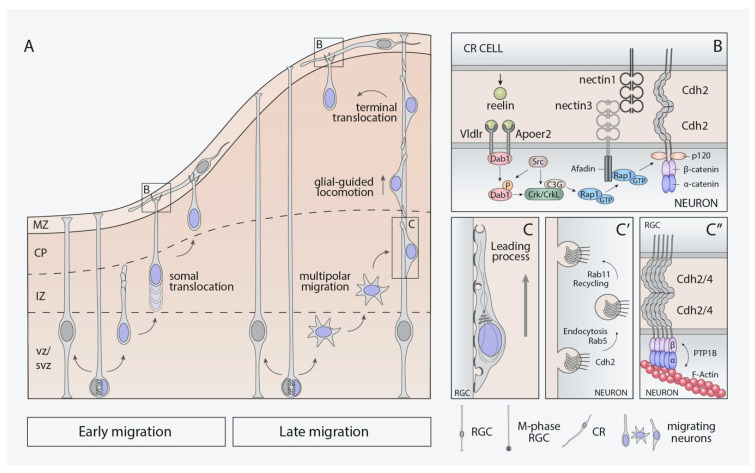
C-CAMs involvement in radial migration during neocortical development. (**A**) radial migration modes used by cortical projection neurons at early and late embryonic ages. (**B**) reelin signaling pathway enhances cell–cell adhesion between Cajal–Retzius cells and migrating neurons via nectin1/3 trans-interaction and Cdh2 recruitment in a Dab1/Rap1-depending way. (**C**) schematics of cadherin vesicle trafficking and nucleokinesis during glia-guided locomotion. (**C′**) Rab5 and Rab11 respectively participate in endocytosis recycling vesicles carrying Cdh2. (**C″**) PTP1B cooperates with α/β-catenins connecting Cdh2/4 to the actin cytoskeleton. Abbreviations: CP, cortical plate; CR, Cajal–Retzius; MZ, marginal zone; RGC, radial glial cells; SVZ, subventricular zone; VZ, ventricular zone.

**Figure 4 cells-10-00118-f004:**
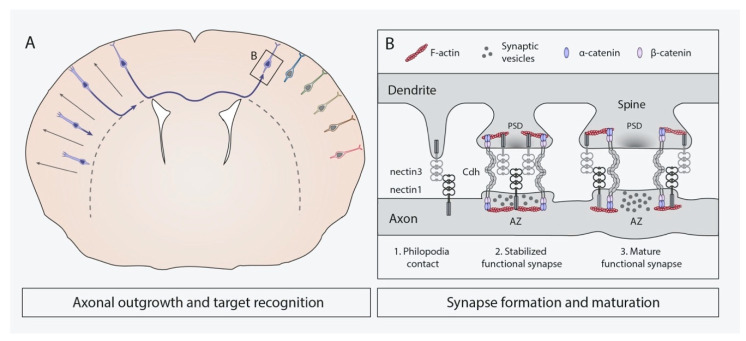
C-CAMs roles in axonal outgrowth, target recognition, and synaptogenesis. (**A**). schematics of axonal pathfinding in the neocortex showing the generation of callosal projections. From left to right: upon polarization postmitotic neurons start extending their axon through the intermediate zone and across the corpus callosum until reaching matching targets (dark blue cells) in the contralateral hemisphere. Different colors represent possible adhesion codes generated by particular expression of C-CAMs. (**B**) schematic representation of the temporal progression of synaptogenesis upon target recognition of the axon. In this example synapse formation begins with interaction of nectins between the axon and philopodia-prottuding dendrites (1), followed by cadherin recruitment to the nascent presynaptic and postsynaptic zones (2), and the progressive assembly of the active zone (AZ) and postsynaptic densities (PSD) to stabilize the synaptic contact, in this case with the formation of a dendritic spine (3).

**Table 1 cells-10-00118-t001:** Interactions among C-CAMs members and described roles during development.

Super Family	Family	Member	Common Interactions	Related Neurodevelopmental Process
Cadherins	Type I Classical Cadherins	Cdh1	Cdh1	Proliferation/differentiation, synaptogenesis
		Cdh2	Cdh2, Cdh4	Proliferation/differentiation, migration, axon guidance, synaptogenesis
		Cdh4	Cdh4	Migration, axon guidance
	Type II Classical Cadherins	Cdh6	Cdh6	Axon guidance
		Cdh8	Cdh8	Proliferation/differentiation, Synaptogenesis
		Cdh11	Cdh11	Synaptogenesis
		Cdh13	Cdh13	Axon guidance
IgCAMs	nectin	nectin1	nectin1, nectin3, nectin4, Necl1, tactile	Migration, axon guidance, synaptogenesis
		nectin3	nectin1, nectin2, nectin3, necl1, Necl2, Necl5, TIGIT	Migration, axon guidance, synaptogenesis
	Necl	Necl1	nectin1, nectin3, Necl1, Necl2, Necl3, Necl4	-
		Necl2	nectin3, Necl1, Necl2, Necl3, Necl2	Axon guidance, Synaptogenesis
		Necl3	Necl1, Necl2	Axon guidance
		Necl4	Necl1, Necl4	-

**Table 2 cells-10-00118-t002:** Overview of the neocortical phenotype of mutant mice for C-CAMs and their related proteins.

Disrupted Adhesive Molecule	Nature of the Genetic Perturbation	Neocorticogenic Processes Affected and Subsequent Phenotypic Alterations	References
Cdh2	D6-Cre-dependent recombination in Cdh2 F/F mice	Disruption of AJs integrity, laminar and RGCs organization, progenitor proliferation and cell cycle exit	[79]
	Emx1-Cre-dependent recombination in Cdh2 F/F mice	Disruption of AJs integrity, cell fate choice and axonal pathfinding; enhanced progenitor proliferation; lamination defects and double cortex formation	[80]
	CaMKII-driven expression of the Cdh2-intracellular domain	Impaired spine pruning and maturation of neocortical excitatory PNs	[85]
afadin	Emx1-Cre-dependent recombination in Mltt4 F/F mice	Disruption of AJs integrity, cell fate choice and axonal pathfinding; enhanced progenitor proliferation; lamination defects and double cortex formation	[80]
	Nestin-Cre-dependent recombination in Mltt4 F/F mice	Disruption of AJs integrity; mislocalization of ependymal cells; hydrocephalus and neonatal death	[84]
αE-catenin	Nestin-Cre dependent recombination in αE-catenin F/F mice	Disruption of AJs integrity; cell cycle shortening, increased proliferation and decreased apoptosis of progenitor cells; cortical dysplasia and hyperplasia	[83]
αN-catenin	Spontaneous mutation leading to αN-catenin gene deletion	Impaired spine pruning and maturation of neocortical excitatory PNs	[85]
β-catenin	D6-Cre-dependent recombination in β-catenin F/F mice	Disruption of AJs integrity, interkinetic nuclear migration, RGCs organization, radial migration of early-born neurons and cell fate choice; decreased progenitor proliferation	[82]
	Nex-Cre-dependent recombination in β-catenin F(ex3)/F(ex3) mice generating a stable and active form of β-catenin	Enhanced synapse and spine stability and density in layer II/III excitatory PNs’ apical and basal dendrites	[86]
	Scnn1a-Tg3-Cre-dependent recombination in β-catenin F(ex3)/F(ex3) mice, generating a stabilized and active form of β-catenin only in layer IV excitatory PNs	Enhanced synapse and spine stability and density in layer II/III excitatory PNs’ basal dendrites	[86]
	CaMKII-Cre-ERT2-dependent recombination in β-catenin F/F mice	Impaired spine pruning and maturation of neocortical excitatory PNs	[85]
Necl2	Knockin mice with the lacZ-neo-cassette replacing part of Necl2 gene sequence, ablating Necl2 expression	Decreased density of thalamic synaptic inputs in PV^+^ interneurons from layer IV of the primary visual neocortex	[87]

**Table 3 cells-10-00118-t003:** Summary of known C-CAMs mutations in different human neurodevelopmental disorders.

C-CAMs		Genetic Marker *	Neurodevelopmental Disorder	Neocortex-Related Phenotype	References
Cdh SF	CDH2	De novo mutations (D353N, D597N, D597Y, D601T, C613W, D627G, Y676C, L855V, L856F)	ID	Yes	[143]
		rs17445840, rs2289664, De novo mutations (N706S, V289I)	TD, OCD	No	[195]
	CDH4	De novo mutations (E451K, A852T, R659P)	MCD	Yes	[196]
	CDH7	rs1444067, rs2850700, rs2658046, rs12970791, rs2850699, rs4455070	BD	No	[191]
	CDH8	De novo mutation (16q21 microdeletion)	ASD, LD	Yes	[186]
	CDH9/CDH10	rs4307059, rs7704909, rs12518194, rs4327572, rs1896731, rs10038113	ASD	No	[185]
	CDH11	rs10500464 rs429065	ASD, ADHD	No	[184]
	CDH13	rs6565113, rs11646411	ADHD	No	[189,190]
		rs8057927	SCZ	No	[192]
	CDH15	De novo mutations (V8L, R60C, R92W, A122V)	ID	No	[193]
		De novo mutation (16q24.3 microdeletion)	ID	Yes	[194]
Necls	NECL2	rs10891819	ADHD	Yes	[188]
		De novo mutations (H246N, Y251S)	ASD	No	[187]
		*De novo* mutations (H246N, Y251S)	ASD	No	[183]
	NECL3	De novo mutation (3p12.1 microdeletion)	ASD	No	[182]

* Single nucleotides polymorphisms (SNPs) are indicated using RefSNP number (rs) and de novo mutations are indicated in the protein sequence.

## Data Availability

Not applicable.

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
