# Peer review of "Stick around: Cell–Cell Adhesion Molecules during Neocortical Development"

_cells, 2021, doi:10.3390/cells10010118_

Round 1

Reviewer 1 Report

This is a well written review on the molecular function of cell-cell adhesion molecules (C-CAMs), classical cadherins and nectins. Biological significance could be expended on. I would recommend to incorporate a table on mouse models and their phenotypes; these should also be mentioned in the text where the specific CAM is discussed.

Human genetic relevance is another important piece of information to be incorporated in this review. This can be presented in a summary table including gene, mutation and the clinical phenotype and reference to the databases where the information was extracted from.

Author Response

We would like to thank the reviewers for their constructive criticisms and suggestions that have helped us to improve the manuscript. We have addressed the concerns raised by the reviewers and we now include additional tables (Tables 2 and 3), we have modified Figure 2 as requested and finally we added a new section related to human genetic relevance of the C-CAMs object of this review. We have used track changes in word to facilitate the identification of our editions.

Below is our point-by point response to the reviewers´ comments:

Reviewer #1:

  • “This is a well written review on the molecular function of cell-cell adhesion molecules (C-CAMs), classical cadherins and nectins. Biological significance could be expended on. I would recommend to incorporate a table on mouse models and their phenotypes; these should also be mentioned in the text where the specific CAM is discussed”

We would like first to thank reviewer #1 for his/her comments. We believe that to generate a table summarizing the different phenotypes of the different mouse models described on the manuscript is a fantastic idea, because it could considerably help to identify similarities and differences among them in a quick manner. Consequently, we have added “Table 2. Overview of the neocortical phenotype of mutant mice for C-CAMs and their related proteins” (Line 540) and cited it in all the different sections of the text where we describe the phenotype of the different mutants. We have added references to the original papers to facilitate the readers the identifications of those works.

  • “Human genetic relevance is another important piece of information to be incorporated in this review. This can be presented in a summary table including gene, mutation and the clinical phenotype and reference to the databases where the information was extracted from.”

We totally agree with reviewer #1 that human genetic relevance of those molecules is a very important aspect that deserves attention in this review, so we would like to apologize for having forgotten to add such relevant information in the previous version of the manuscript. We have added a new section to the manuscript entitled “3.4. Relevance of classical cadherins, nectins and Necls in human neurodevelopmental disorders” (line 1128), to summarize the findings related to the existence of human mutations for these proteins and its association with diverse neurodevelopmental disorders. We have also tried to recapitulate the available data about mutations in those proteins and particular neocortical alterations that could be linked to neuropsychiatric diseases. Additionally, we included “Table 3. Summary of known C-CAMs mutations in different human neurodevelopmental disorders” (line 1269), as suggested by the reviewer, including gene name, known mutations and the clinical phenotype properly refereeing to the original works. Likewise, we have modified the discussion to include this new data.

Reviewer 2 Report

The manuscript “Stick around: cell-cell adhesion molecules during neocortical development” by de Agustín-Durán et al. is well written review and well illustrated. I do not have main criticisms.

I have only a few minor suggestions on the manuscript.

Point1. I would just like to suggest to the authors to specify the CAM molecules involved in brain disorders in humans. It could be appropriate to add a table or a column in the table 1 where show these information.

Point2. The authors well describe the points in the figures and figure legends. However in figure 2, I suggest to improve the cartoon indicating the expanded region of the cortex.

1.0.0.20 1.0.0.20

Author Response

We would like to thank the reviewers for their constructive criticisms and suggestions that have helped us to improve the manuscript. We have addressed the concerns raised by the reviewers and we now include additional tables (Tables 2 and 3), we have modified Figure 2 as requested and finally we added a new section related to human genetic relevance of the C-CAMs object of this review. We have used track changes in word to facilitate the identification of our editions.

Below is our point-by point response to the reviewers´ comments:

Reviewer #2:

“The manuscript “Stick around: cell-cell adhesion molecules during neocortical development” by de Agustín-Durán et al. is well written review and well illustrated. I do not have main criticisms. I have only a few minor suggestions on the manuscript”

  • I would just like to suggest to the authors to specify the CAM molecules involved in brain disorders in humans. It could be appropriate to add a table or a column in the table 1 where show these information.

First of all, we want to thank reviewer #2 for his/her comments. We totally agree with this suggestion in the same terms that we do with the one made by reviewer #1. As mentioned in the comments to this reviewer, we have added a new section to the manuscript entitled “3.4. Relevance of classical cadherins, nectins and Necls in human neurodevelopmental disorders” (line 1128), to summarize the findings related to the existence of human mutations for these proteins and its association with diverse neurodevelopmental disorders. We have also summarized the available data about mutations in those proteins and particular neocortical alterations that could be linked to neurological diseases. Additionally, we included “Table 3. Summary of known C-CAMs mutations in different human neurodevelopmental disorders” (line 1269), summarizing the known information about human mutations and related neurodevelopmental disorders. Likewise, we have modified the discussion to include this new data.

  • The authors well describe the points in the figures and figure legends. However, in figure 2, I suggest to improve the cartoon indicating the expanded region of the cortex.

We would like to apologize if something in the previous figure was unclear for Reviewer #2. We have tried to improve it modifying the central cartoon to clarify the general increase in proliferation and cortical expansion, since early ages, together with cell and non-cell autonomous migration defects that cause double cortex formation as described in Gil-Sanz et al., 2014. As a consequence of those defects, the animals present ectopic white matter between the normocortex and the heterotopic cortex. The defects in proliferation are more evident in the enlarged heterotopic cortex. New figure 2 is added (line 606). We hope this new figure could satisfy the concerns of reviewer #2.

Reviewer 3 Report

De Agustin-Duran and his colleagues have written a complete review on two class of cell-cell adhesion molecules contributing to neocortical development: cadherins and nectins.

This manuscript is well written and the role of these two families of cell adhesion molecules in neocorticogenesis and in neuronal migration is well described. Furthermore, these authors have reported how cadherins and nectins participate to synapse formation and cortical remodeling.

Author Response

We would like to thank the reviewers for their constructive criticisms and suggestions that have helped us to improve the manuscript. We have addressed the concerns raised by the reviewers and we now include additional tables (Tables 2 and 3), we have modified Figure 2 as requested and finally we added a new section related to human genetic relevance of the C-CAMs object of this review. We have used track changes in word to facilitate the identification of our editions.

Below is our point-by point response to the reviewers´ comments:

Reviewer #3

“De Agustin-Duran and his colleagues have written a complete review on two class of cell-cell adhesion molecules contributing to neocortical development: cadherins and nectins.

This manuscript is well written and the role of these two families of cell adhesion molecules in neocorticogenesis and in neuronal migration is well described. Furthermore, these authors have reported how cadherins and nectins participate to synapse formation and cortical remodeling”

We want to thank reviewer #3 for his/her comments. We are really glad he/she liked our work and we hope that the new information added to the manuscript, related with the concerns of the other reviewers, would also be to his/her liking.